# Direct Estimation of Differences in Causal Graphs

**Yuhao Wang**
Lab for Information & Decision Systems
and Institute for Data, Systems and Society
Massachusetts Institute of Technology
Cambridge, MA 02139
yuhaow@mit.edu

**Chandler Squires**
Lab for Information & Decision Systems
and Institute for Data, Systems and Society
Massachusetts Institute of Technology
Cambridge, MA 02139
csquires@mit.edu

**Anastasiya Belyaeva**
Lab for Information & Decision Systems
and Institute for Data, Systems and Society
Massachusetts Institute of Technology
Cambridge, MA 02139
belyaeva@mit.edu

**Caroline Uhler**
Lab for Information & Decision Systems
and Institute for Data, Systems and Society
Massachusetts Institute of Technology
Cambridge, MA 02139
cuhler@mit.edu

## Abstract

We consider the problem of estimating the differences between two causal directed acyclic graph (DAG) models with a shared topological order given i.i.d. samples from each model. This is of interest for example in genomics, where changes in the structure or edge weights of the underlying causal graphs reflect alterations in the gene regulatory networks. We here provide the first provably consistent method for directly estimating the differences in a pair of causal DAGs without separately learning two possibly large and dense DAG models and computing their difference. Our two-step algorithm first uses invariance tests between regression coefficients of the two data sets to estimate the skeleton of the difference graph and then orients some of the edges using invariance tests between regression residual variances. We demonstrate the properties of our method through a simulation study and apply it to the analysis of gene expression data from ovarian cancer and during T-cell activation.

## 1  Introduction

Directed acyclic graph (DAG) models, also known as Bayesian networks, are widely used to model causal relationships in complex systems. Learning the *causal DAG* from observations on the nodes is an important problem across disciplines [8, 25, 30, 36]. A variety of causal inference algorithms based on observational data have been developed, including the prominent PC [36] and GES [20] algorithms, among others [35, 38]. However, these methods require strong assumptions [39]; in particular, theoretical analysis of the PC [14] and GES [23, 40] algorithms have shown that these methods are usually not consistent in the high-dimensional setting, i.e. when the number of nodes is of the same order or exceeds the number of samples, unless highly restrictive assumptions on the sparsity and/or the maximum degree of the underlying DAG are made.

The presence of high degree hub nodes is a well-known feature of many networks [2, 3], thereby limiting the direct applicability of causal inference algorithms. However, in many applications, the end goal is not to recover the full causal DAG but to detect changes in the causal relations between two related networks. For example, in the analysis of EEG signals it is of interest to detect neurons or different brain regions that interact differently when the subject is performing different activities [31]; in biological pathways genes may control different sets of target genes under different

cellular contexts or disease states [11, 28]. Due to recent technological advances that allow the collection of large-scale EEG or single-cell gene expression data sets in different contexts there is a growing need for methods that can accurately identify differences in the underlying regulatory networks and thereby provide key insights into the underlying system [11, 28]. The limitations of causal inference algorithms for accurately learning large causal networks with hub nodes and the fact that the difference of two related networks is often sparse call for methods that directly learn the difference of two causal networks without having to estimate each network separately.

The complimentary problem to learning the difference of two DAG models is the problem of inferring the causal structure that is *invariant* across different environments. Algorithms for this problem have been developed in recent literature [9, 27, 42]. However, note that the difference DAG can only be inferred from the invariant causal structure if the two DAGs are known. The problem of learning the difference between two networks has been considered previously in the *undirected* setting [17, 18, 43]. However, the undirected setting is often insufficient: only a causal (i.e., directed) network provides insights into the effects of interventions such as knocking out a particular gene. In this paper we provide to our knowledge the first provably consistent method for directly inferring the differences between pairs of causal DAG models that does not require learning each model separately.

The remainder of this paper is structured as follows: In Section 2, we set up the notation and review related work. In Section 3, we present our algorithm for directly estimating the difference of causal relationships and in Section 4, we provide consistency guarantees for our algorithm. In Section 5, we evaluate the performance of our algorithm on both simulated and real datasets including gene expression data from ovarian cancer and T cell activation.

## 2   Preliminaries and Related Work

Let $\mathcal{G} = ([p], A)$ be a DAG with node set $[p] := \{1, \cdots, p\}$ and arrow set $A$. Without loss of generality we label the nodes such that if $i \to j$ in $\mathcal{G}$ then $i < j$ (also known as *topological ordering*). To each node $i$ we associate a random variable $X_i$ and let $\mathbb{P}$ be a joint distribution over the random vector $X = (X_1, \cdots, X_p)^T$. In this paper, we consider the setting where the causal DAG model is given by a *linear structural equation model (SEM) with Gaussian noise*, i.e.,

$$X = B^T X + \epsilon$$

where $B$ (the *autoregressive matrix*) is strictly upper triangular consisting of the edge weights of $\mathcal{G}$, i.e., $B_{ij} \neq 0$ if and only if $i \to j$ in $\mathcal{G}$, and the noise $\epsilon \sim \mathcal{N}(0, \Omega)$ with $\Omega := \mathrm{diag}(\sigma_1^2, \cdots, \sigma_p^2)$, i.e., there are no latent confounders. Denoting by $\Sigma$ the covariance matrix of $X$ and by $\Theta$ its inverse (i.e., the *precision matrix*), a short computation yields $\Theta = (I - B)\Omega^{-1}(I - B)^T$, and hence

$$\Theta_{ij} = -\sigma_j^{-2} B_{ij} + \sum_{k>j} \sigma_k^{-2} B_{ik} B_{jk}, \ \ \forall i \neq j \quad \text{and} \quad \Theta_{ii} = \sigma_i^{-2} + \sum_{j>i} \sigma_j^{-2} B_{ij}^2, \ \ \forall i \in [p]. \quad (1)$$

This shows that the support of $\Theta$ is given by the *moral graph* of $\mathcal{G}$, obtained by adding an edge between pairs of nodes that have a common child and removing all edge orientations. By the *causal Markov assumption*, which we assume throughout, the missing edges in the moral graph encode a subset of the conditional independence (CI) relations implied by a DAG model on $\mathcal{G}$; the complete set of CI relations is given by the *d-separations* that hold in $\mathcal{G}$ [15][Section 3.2.2]; i.e., $X_i \perp\!\!\!\perp X_j \mid X_S$ in $\mathbb{P}$ whenever nodes $i$ and $j$ are d-separated in $\mathcal{G}$ given a set $S \subseteq [p] \setminus \{i, j\}$. The *faithfulness assumption* is the assertion that all CI relations entailed by $\mathbb{P}$ are implied by d-separation in $\mathcal{G}$.

A standard approach for causal inference is to first infer CI relations from the observations on the nodes of $\mathcal{G}$ and then to use the CI relations to learn the DAG structure. However, several DAGs can encode the same CI relations and therefore, $\mathcal{G}$ can only be identified up to an equivalence class of DAGs, known as the *Markov equivalence class* (MEC) of $\mathcal{G}$, which we denote by $[\mathcal{G}]$. In [41], the author gave a graphical characterization of the members of $[\mathcal{G}]$; namely, two DAGs are in the same MEC if and only if they have the same *skeleton* (i.e., underlying undirected graph) and the same *v-structures* (i.e., induced subgraphs of the form $i \to j \leftarrow k$). $[\mathcal{G}]$ can be represented combinatorially by a partially directed graph with skeleton $\mathcal{G}$ and an arrow for those edges in $\mathcal{G}$ that have the same orientation in all members of $[\mathcal{G}]$. This is known as the *CP-DAG* (or *essential graph*) of $\mathcal{G}$ [1].

Various algorithms have been developed for learning the MEC of $\mathcal{G}$ given observational data on the nodes, most notably the prominent GES [20] and PC algorithms [36]. While GES is a score-based

approach that greedily optimizes a score such as the BIC (Bayesian Information Criterion) over the space of MECs, the PC algorithm views causal inference as a constraint satisfaction problem with the constraints being the CI relations. In a two-stage approach, the PC algorithm first learns the skeleton of the underlying DAG and then determines the v-structures, both from the inferred CI relations. GES and the PC algorithms are provably consistent, meaning they output the correct MEC given an infinite amount of data, under the faithfulness assumption. Unfortunately, this assumption is very sensitive to hypothesis testing errors for inferring CI relations from data and violations are frequent especially in non-sparse graphs [39]. If the noise variables in a linear SEM with additive noise are non-Gaussian, the full causal DAG can be identified (as opposed to just its MEC) [33], for example using the prominent LiNGAM algorithm [33]. Non-Gaussianity and sparsity of the underlying graph in the high-dimensional setting are crucial for consistency of LiNGAM.

In this paper, we develop a two-stage approach, similar to the PC algorithm, for directly learning the difference between two linear SEMs with additive Gaussian noise on the DAGs $\mathcal{G}$ and $\mathcal{H}$. Note that naive algorithms that separately estimate $[\mathcal{G}]$ and $[\mathcal{H}]$ and take their differences can only identify edges that appeared/disappeared and cannot identify changes in edge weights (since the DAGs are not identifiable). Our algorithm overcomes this limitation. In addition, we show in Section 4 that instead of requiring the restrictive faithfulness assumption on both DAGs $\mathcal{G}$ and $\mathcal{H}$, consistency of our algorithm only requires assumptions on the (usually) smaller and sparser network of differences.

Let $\mathbb{P}^{(1)}$ and $\mathbb{P}^{(2)}$ be a pair of linear SEMs with Gaussian noise defined by $(B^{(1)}, \epsilon^{(1)})$ and $(B^{(2)}, \epsilon^{(2)})$. Throughout, we make the simplifying assumption that both $B^{(1)}$ and $B^{(2)}$ are strictly upper triangular, i.e., that the underlying DAGs $\mathcal{G}^{(1)}$ and $\mathcal{G}^{(2)}$ share the same topological order. This assumption is reasonable for example in applications to genomics, since genetic interactions may appear or disappear, change edge weights, but generally do not change directions. For example, in biological pathways an upstream gene does not generally become a downstream gene in different conditions. Hence $B^{(1)} - B^{(2)}$ is also strictly upper triangular and we define the *difference-DAG* (*D-DAG*) of the two models by $\Delta := ([p], A_\Delta)$ with $i \to j \in A_\Delta$ if and only if $B_{ij}^{(1)} \neq B_{ij}^{(2)}$; i.e., an edge $i \to j$ in $\Delta$ represents a change in the causal effect of $i$ on $j$, including changes in the presence/absence of an effect as well as changes in edge weight. Given i.i.d. samples from $\mathbb{P}^{(1)}$ and $\mathbb{P}^{(2)}$, our goal is to infer $\Delta$. Just like estimating a single causal DAG model, the D-DAG $\Delta$ is in general not completely identifiable, in which case we wish to identify the skeleton $\bar{\Delta}$ as well as a subset of arrows $\tilde{A}_\Delta$.

A simpler task is learning differences of undirected graphical models. Let $\Theta^{(1)}$ and $\Theta^{(2)}$ denote the precision matrices corresponding to $\mathbb{P}^{(1)}$ and $\mathbb{P}^{(2)}$. The support of $\Theta^{(k)}$ consists of the edges in the undirected graph (UG) models corresponding to $\mathbb{P}^{(k)}$. We define the *difference-UG* (*D-UG*) by $\Delta_\Theta := ([p], E_{\Delta_\Theta})$, with $i - j \in E_{\Delta_\Theta}$ if and only if $\Theta_{ij}^{(1)} \neq \Theta_{ij}^{(2)}$ for $i \neq j$. Two recent methods that directly learn the difference of two UG models are KLIEP [18] and DPM [43]; for a review and comparison of these methods see [17]. These methods can be used as a first step towards estimating the D-DAG $\Delta$: under genericity assumptions, the formulae for $\Theta_{ij}$ in (1) imply that if $B_{ij}^{(1)} \neq B_{ij}^{(2)}$ then $\Theta_{ij}^{(1)} \neq \Theta_{ij}^{(2)}$. Hence, the skeleton of $\Delta$ is a subgraph of $\Delta_\Theta$, i.e., $\bar{\Delta} \subseteq \Delta_\Theta$. In the following section we present our algorithm showing how to obtain $\bar{\Delta}$ and determine some of the edge directions in $\Delta$. We end this section with a piece of notation needed for introducing our algorithm; we define the *set of changed nodes* to be $S_\Theta := \{i \mid \exists j \in [p] \text{ such that } \Theta_{i,j}^{(1)} \neq \Theta_{i,j}^{(2)}\}$.

## 3 Difference Causal Inference Algorithm

In Algorithm 1 we present our *Difference Causal Inference* (*DCI*) algorithm for directly learning the difference between two linear SEMs with additive Gaussian noise given i.i.d. samples from each model. Our algorithm consists of a two-step approach similar to the PC algorithm. The first step,

---

**Algorithm 1** Difference Causal Inference (DCI) algorithm

---

**Input:** Sample data $\hat{X}^{(1)}, \hat{X}^{(2)}$.
**Output:** Estimated skeleton $\bar{\Delta}$ and arrows $\tilde{A}_\Theta$ of the D-DAG $\Delta$.

Estimate the D-UG $\Delta_\Theta$ and $S_\Theta$; use Algorithm 2 to estimate $\bar{\Delta}$; use Algorithm 3 to estimate $\tilde{A}_\Delta$.

---

**Algorithm 2** Estimating skeleton of the D-DAG

---

**Input:** Sample data $\hat{X}^{(1)}$, $\hat{X}^{(2)}$, estimated D-UG $\Delta_\Theta$, estimated set of changed nodes $S_\Theta$.
**Output:** Estimated skeleton $\bar{\Delta}$.

Set $\bar{\Delta} := \Delta_\Theta$;
**for** each edge $i - j$ in $\bar{\Delta}$ **do**
    If $\exists S \subseteq S_\Theta \setminus \{i, j\}$ such that $\beta_{i,j|S}^{(k)}$ is invariant across $k = \{1, 2\}$, delete $i - j$ in $\bar{\Delta}$ and continue
    to the next edge. Otherwise, continue.
**end for**

---

---

**Algorithm 3** Directing edges in the D-DAG

---

**Input:** Sample data $\hat{X}^{(1)}$, $\hat{X}^{(2)}$, estimated set of changed nodes $S_\Theta$, estimated skeleton $\bar{\Delta}$.
**Output:** Estimated set of arrows $\tilde{A}_\Delta$.

Set $\tilde{A}_\Delta := \emptyset$;
**for** each node $j$ incident to at least one undirected edge in $\bar{\Delta}$ **do**
    If $\exists S \subseteq S_\Theta \setminus \{j\}$ such that $\sigma_{j|S}^{(k)}$ is invariant across $k = \{1, 2\}$, add $i \rightarrow j$ to $\tilde{A}_\Delta$ for all $i \in S$,
    and add $j \rightarrow i$ to $\tilde{A}_\Delta$ for all $i \notin S$ and continue to the next node. Otherwise, continue.
**end for**
Orient as many undirected edges as possible via graph traversal using the following rule:
    Orient $i - j$ into $i \rightarrow j$ whenever there is a chain $i \rightarrow \ell_1 \rightarrow \cdots \rightarrow \ell_t \rightarrow j$.

---

described in Algorithm 2, estimates the skeleton of the D-DAG by removing edges one-by-one. Algorithm 2 takes $\Delta_\Theta$ and $S_\Theta$ as input. In the high-dimensional setting, KLIEP can be used to estimate $\Delta_\Theta$ and $S_\Theta$. For completeness, in the Supplementary Material we also provide a constraint-based method that consistently estimates $\Delta_\Theta$ and $S_\Theta$ in the low-dimensional setting for general additive noise models. Finally, $\Delta_\Theta$ can also simply be chosen to be the complete graph with $S_\Theta = [p]$. These different initiations of Algorithm 2 are compared via simulations in Section 5. The second step of DCI, described in Algorithm 3, infers some of the edge directions in the D-DAG. While the PC algorithm uses CI tests based on the partial correlations for inferring the skeleton and for determining edge orientations, DCI tests the invariance of certain *regression coefficients* across the two data sets in the first step and the invariance of certain *regression residual variances* in the second step. These are similar to the regression invariances used in [9] and are introduced in the following definitions.

**Definition 3.1.** Given $i, j \in [p]$ and $S \subseteq [p] \setminus \{i, j\}$, let $M := \{i\} \cup S$ and let $\beta_M^{(k)}$ be the best linear predictor of $X_j^{(k)}$ given $X_M^{(k)}$, i.e., the minimizer of $\mathbb{E}[(X_j^{(k)} - (\beta_M^{(k)})^T X_M^{(k)})^2]$. We define the *regression coefficient* $\beta_{i,j|S}^{(k)}$ to be the entry in $\beta_M^{(k)}$ corresponding to $i$.

**Definition 3.2.** For $j \in [p]$ and $S \subseteq [p] \setminus \{j\}$, we define $(\sigma_{j|S}^{(k)})^2$ to be the *variance of the regression residual* when regressing $X_j^{(k)}$ onto the random vector $X_S^{(k)}$.

Note that in general $\beta_{i,j|S}^{(k)} \neq \beta_{j,i|S}^{(k)}$. Each entry in $B^{(k)}$ can be interpreted as a regression coefficient, namely $B_{ij}^{(k)} = \beta_{i,j|(\text{Pa}^{(k)}(j) \setminus \{i\})}^{(k)}$, where $\text{Pa}^{(k)}(j)$ denotes the parents of node $j$ in $\mathcal{G}^{(k)}$. Thus, when $B_{ij}^{(1)} = B_{ij}^{(2)}$, then there exists a set $S$ such that $\beta_{i,j|S}^{(k)}$ stays invariant across $k = \{1, 2\}$. This motivates using invariances between regression coefficients to determine the skeleton of the D-DAG. For orienting edges, observe that when $\sigma_j^{(k)}$ stays invariant across two conditions, $\sigma_{j|S}^{(k)}$ would also stay invariant if $S$ is chosen such that $S = \text{Pa}^{(1)}(j) \cup \text{Pa}^{(2)}(j)$. This motivates using invariances of residual variances to discover the parents of node $j$ and assign orientations afterwards. Similar to [9] we use hypothesis tests based on the F-test for testing the invariance between regression coefficients and residual variances. See the Supplementary Material for details regarding the construction of these hypothesis tests, the derivation of their asymptotic distribution, and an example outlining the difference of this approach to [9] for invariant structure learning.

**Example 3.3.** *We end this section with a 4-node example showing how the DCI algorithm works. Let $B^{(1)}$ and $B^{(2)}$ be the autoregressive matrices defined by the edge weights of $\mathcal{G}^{(1)}$ and $\mathcal{G}^{(2)}$ and let the noise variances satisfy the following invariances:*

$$\sigma_1^{(1)} \neq \sigma_1^{(2)}, \quad \sigma_3^{(1)} = \sigma_3^{(2)},$$
$$\sigma_2^{(1)} = \sigma_2^{(2)}, \quad \sigma_4^{(1)} \neq \sigma_4^{(2)};$$

*Initiating Algorithm 2 with $\Delta_\Theta$ being the complete graph and $S_\Theta = [4]$, the output of the DCI algorithm is shown above.*

## 4 Consistency of DCI

The DCI algorithm is *consistent* if it outputs a partially oriented graph $\hat{\Delta}$ that has the same skeleton as the true D-DAG and whose oriented edges are all correctly oriented. Just as methods for estimating a single DAG require assumptions on the underlying model (e.g. the faithfulness assumption) to ensure consistency, our method for estimating the D-DAG requires assumptions on relationships between the two underlying models. To define these assumptions it is helpful to view $(\sigma_j^{(k)})_{j\in[p]}$ and the non-zero entries $(B_{ij}^{(k)})_{(i,j)\in A^{(k)}}$ as *variables* or *indeterminates* and each entry of $\Theta^{(k)}$ as a *rational function*, i.e., a fraction of two *polynomials* in the variables $B_{ij}^{(k)}$ and $\sigma_j^{(k)}$ as defined in (1). Using Schur complements one can then similarly express $\beta_{v,w|S}^{(k)}$ and $(\sigma_{w|S}^{(k)})^2$ as a rational function in the entries of $\Theta^{(k)}$ and hence as a rational function in the variables $(B_{ij}^{(k)})_{(i,j)\in A^{(k)}}$ and $(\sigma_j^{(k)})_{j\in[p]}$. The exact formulae are given in the Supplementary Material.

Clearly, if $B_{ij}^{(1)} = B_{ij}^{(2)} \; \forall(i,j)$ and $\sigma_j^{(1)} = \sigma_j^{(2)} \; \forall j \in [p]$, then $\beta_{v,w|S}^{(1)} = \beta_{v,w|S}^{(2)}$ and $\sigma_{w|S}^{(1)} = \sigma_{w|S}^{(2)}$ for all $v, w, S$. For consistency of the DCI algorithm we assume that the converse is true as well, namely that *differences* in $B_{ij}$ and $\sigma_j$ in the two distributions are not "cancelled out" by changes in other variables and result in *differences* in the regression coefficients and regression residual variances. This allows us to deduce invariance patterns of the autoregressive matrix $B^{(k)}$ from invariance patterns of the regression coefficients and residual variances, and hence differences of the two causal DAGs.[1]

**Assumption 4.1.** *For any choice of $i, j \in S_\Theta$, if $B_{ij}^{(1)} \neq B_{ij}^{(2)}$ then for all $S \subseteq S_\Theta \setminus \{i, j\}$ it holds that*

$$\beta_{i,j|S}^{(1)} \neq \beta_{i,j|S}^{(2)} \text{ and } \beta_{j,i|S}^{(1)} \neq \beta_{j,i|S}^{(2)}.$$

**Assumption 4.2.** *For any choice of $i, j \in S_\Theta$ it holds that*

1. *if $B_{ij}^{(1)} \neq B_{ij}^{(2)}$, then $\forall S \subseteq S_\Theta \setminus \{i, j\}$, $\sigma_{j|S}^{(1)} \neq \sigma_{j|S}^{(2)}$ and $\sigma_{i|S\cup\{j\}}^{(1)} \neq \sigma_{i|S\cup\{j\}}^{(2)}$.*
2. *if $\sigma_j^{(1)} \neq \sigma_j^{(2)}$, then $\sigma_{j|S}^{(1)} \neq \sigma_{j|S}^{(2)}$ for all $S \subseteq S_\Theta \setminus \{j\}$.*

Assumption 4.1 ensures that the skeleton of the D-DAG is inferred correctly, whereas Assumption 4.2 ensures that the arrows returned by the DCI algorithm are oriented correctly. These assumptions are the equivalent of the *adjacency-faithfulness* and *orientation-faithfulness* assumptions that ensure consistency of the PC algorithm for estimating the MEC of a causal DAG [29].

We now provide our main results, namely consistency of the DCI algorithm. For simplicity we here discuss the consistency guarantees when Algorithm 2 is initialized with $\Delta_\Theta$ being the complete graph and $S_\Theta = [p]$. However, in practice we recommend initialization using KLIEP (see also Section 5) to avoid performing an unnecessarily large number of conditional independence tests. The consistency guarantees for such an initialization including a method for learning the D-DAG in general additive noise models (that are not necessarily Gaussian) is provided in the Supplementary Material.

**Theorem 4.3.** *Given Assumption 4.1, Algorithm 2 is consistent in estimating the skeleton of the D-DAG $\Delta$.*

The proof is given in the Supplementary Material. The main ingredient is showing that if $B_{ij}^{(1)} = B_{ij}^{(2)}$, then there exists a conditioning set $S \subseteq S_\Theta \setminus \{i, j\}$ such that $\beta_{i,j|S}^{(1)} = \beta_{i,j|S}^{(2)}$, namely the parents of node $j$ in both DAGs excluding node $i$. Next, we provide consistency guarantees for Algorithm 3.

**Theorem 4.4.** *Given Assumption 4.2, all arrows $\tilde{A}_\Delta$ output by Algorithm 3 are correctly oriented. In particular, if $\sigma_i^{(k)}$ is invariant across $k = \{1, 2\}$, then all edges adjacent to $i$ are oriented.*

Similar to the proof of Theorem 4.3, the proof follows by interpreting the rational functions corresponding to regression residual variances in terms of d-connecting paths in $\mathcal{G}^{(k)}$ and is given in the Supplementary Material. It is important to note that as a direct corollary to Theorem 4.4 we obtain sufficient conditions for full identifiability of the D-DAG (i.e., all arrows) using the DCI algorithm.

**Corollary 4.5.** *Given Assumptions 4.1 and 4.2, and assuming that the error variances are the same across the two distributions, i.e. $\Omega^{(1)} = \Omega^{(2)}$, the DCI algorithm outputs the D-DAG $\Delta$.*

In addition, we conjecture that Algorithm 3 is *complete*, i.e., that it directs all edges that are identifiable in the D-DAG. We end this section with two remarks, namely regarding the sample complexity of the DCI algorithm and an evaluation of how restrictive Assumptions 4.1 and 4.2 are.

**Remark 4.6** (Sample complexity of DCI). *For constraint-based methods such as the PC or DCI algorithms, the sample complexity is determined by the number of hypothesis tests performed by the algorithm [14]. In the high-dimensional setting, the number of hypothesis tests performed by the PC algorithm scales as $\mathcal{O}(p^s)$, where $p$ is the number of nodes and $s$ is the maximum degree of the DAG, thereby implying severe restrictions on the sparsity of the DAG given a reasonable sample size. Meanwhile, the number of hypothesis tests performed by the DCI algorithm scales as $\mathcal{O}(|\Delta_\Theta|2^{|S_\Theta|-1})$ and hence does not depend on the degree of the two DAGs. Therefore, even if the two DAGs $\mathcal{G}^{(1)}$ and $\mathcal{G}^{(2)}$ are high-dimensional and highly connected, the DCI algorithm is consistent and has a better sample complexity (as compared to estimating two DAGs separately) as long as the differences between $\mathcal{G}^{(1)}$ and $\mathcal{G}^{(2)}$ are sparse, i.e., $|S_\Theta|$ is small compared to $p$ and $s$.* $\qquad\square$

**Remark 4.7** (Strength of Assumptions 4.1 and 4.2). *Since faithfulness, a standard assumption for consistency of causal inference algorithms to estimate an MEC, is known to be restrictive [39], it is of interest to compare Assumptions 4.1 and 4.2 to the faithfulness assumption of $\mathbb{P}^{(k)}$ with respect to $\mathcal{G}^{(k)}$ for $k \in \{1, 2\}$. In the Supplementary Material we provide examples showing that Assumptions 4.1 and 4.2 do not imply the faithfulness assumption on the two distributions and vice-versa. However, in the finite sample regime we conjecture Assumptions 4.1 and 4.2 to be weaker than the faithfulness assumption: violations of faithfulness as well as of Assumptions 4.1 and 4.2 correspond to points that are close to conditional independence hypersurfaces [39]. The number of these hypersurfaces (and hence the number of violations) increases in $s$ for the faithfulness assumption and in $S_\Theta$ for Assumptions 4.1 and 4.2. Hence if the two DAGs $\mathcal{G}^{(1)}$ and $\mathcal{G}^{(2)}$ are large and complex while having a sparse difference, then $S_\Theta \ll s$. See the Supplementary Material for more details.* $\qquad\square$

## 5 Evaluation

In this section, we compare our DCI algorithm with PC and GES on both synthetic and real data. The code utilized for the following experiments can be found at `https://github.com/csquires/dci`.

### 5.1 Synthetic data

We analyze the performance of our algorithm in both, the low- and high-dimensional setting. For both settings we generated 100 realizations of pairs of upper-triangular SEMs $(B^{(1)}, \epsilon^{(1)})$ and $(B^{(2)}, \epsilon^{(2)})$. For $B^{(1)}$, the graphical structure was generated using an Erdös-Renyi model with expected neighbourhood size $s$, on $p$ nodes and $n$ samples. The edge weights were uniformly drawn from $[-1, -0.25] \cup [0.25, 1]$ to ensure that they were bounded away from zero. $B^{(2)}$ was then generated from $B^{(1)}$ by adding and removing edges with probability 0.1, i.e.,

$$B_{ij}^{(2)} \overset{\text{i.i.d.}}{\sim} \text{Ber}(0.9) \cdot B_{ij}^{(1)} \text{ if } B_{ij}^{(1)} \neq 0, \quad B_{ij}^{(2)} \overset{\text{i.i.d.}}{\sim} \text{Ber}(0.1) \cdot \text{Unif}([-1, -.25] \cup [.25, 1]) \text{ if } B_{ij}^{(1)} = 0$$

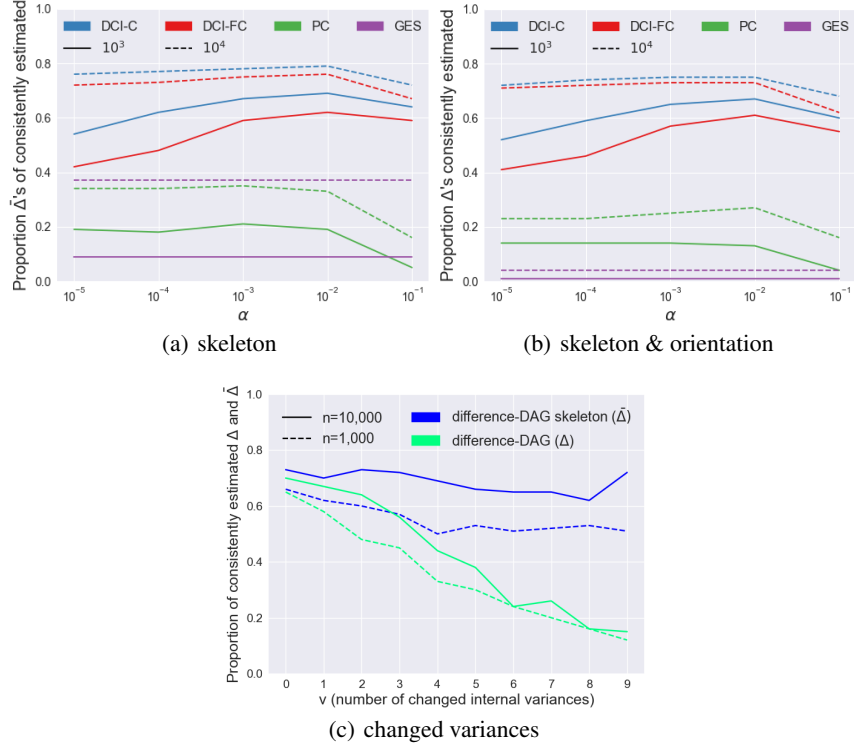

(a) skeleton  (b) skeleton & orientation

(c) changed variances

Figure 1: Proportion of consistently estimated D-DAGs for 100 realizations per setting with $p = 10$ nodes and sample size $n$. Figures (a) and (b) show the proportion of consistently estimated D-DAGs when considering just the skeleton ($\bar{\Delta}$) and both skeleton and edge orientations ($\Delta$), respectively; $\alpha$ is the significance level used for the hypothesis tests in the algorithms. Figure $(c)$ shows the proportion of consistent estimates with respect to the number of changes in internal node variances $v$.

Note that while the DCI algorithm is able to identify changes in edge weights, we only generated DAG models that differ by edge insertions and deletions. This is to provide a fair comparison to the naive approach, where we separately estimate the two DAGs $\mathcal{G}^{(1)}$ and $\mathcal{G}^{(2)}$ and then take their difference, since this approach can only identify insertions and deletions of edges.

In Figure 1 we analyzed how the performance of the DCI algorithm changes over different choices of significance levels $\alpha$. The simulations were performed on graphs with $p = 10$ nodes, neighborhood size of $s = 3$ and sample size $n \in \{10^3, 10^4\}$. For Figure 1 (a) and (b) we set $\epsilon^{(1)}, \epsilon^{(2)} \sim \mathcal{N}(0, \mathbf{1}_p)$, which by Corollary 4.5 ensures that the D-DAG $\Delta$ is fully identifiable. We compared the performance of DCI to the naive approach, where we separately estimated the two DAGs $\mathcal{G}^{(1)}$ and $\mathcal{G}^{(2)}$ and then took their difference. For separate estimation we used the prominent PC and GES algorithms tailored to the Gaussian setting. Since KLIEP requires an additional tuning parameter, to understand how $\alpha$ influences the performance of the DCI algorithm, we here only analyzed initializations in the fully connected graph (DCI-FC) and using the constraint-based method described in the Supplementary Material (DCI-C). Both initializations provide a provably consistent algorithm. Figure 1 (a) and (b) show the proportion of consistently estimated D-DAGs by just considering the skeleton ($\bar{\Delta}$) and both skeleton and orientations ($\Delta$), respectively. For PC and GES, we considered the set of edges that appeared in one estimated skeleton but disappeared in the other as the estimated skeleton of the D-DAG $\bar{\Delta}$. In determining orientations, we considered the arrows that were directed in one estimated CP-DAG but disappeared in the other as the estimated set of directed arrows. Since the main purpose of this low-dimensional simulation study is to validate our theoretical findings, we used the exact recovery rate as evaluation criterion. In line with our theoretical findings, both variants of the DCI algorithm outperformed taking differences after separate estimation. Figure 1 (a) and (b) also show that the PC algorithm outperformed GES, which is unexpected given previous results showing that GES usually has a higher exact recovery rate than the PC algorithm for estimating a single DAG. This is due to the fact that while the PC algorithm usually estimates less DAGs correctly, the incorrectly

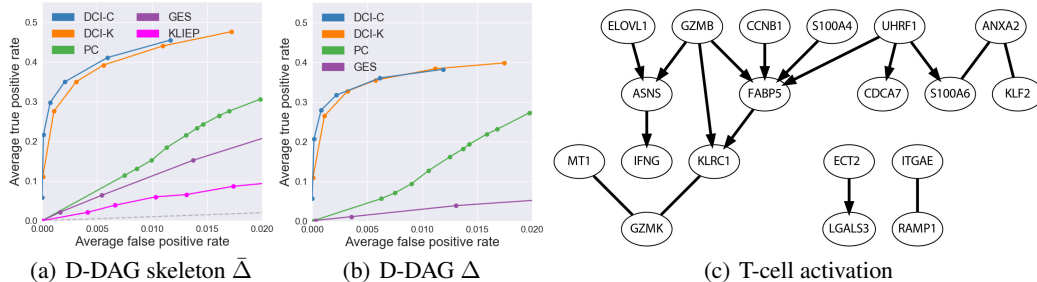

| (a) D-DAG skeleton $\bar{\Delta}$ | (b) D-DAG $\Delta$ | (c) T-cell activation |

Figure 2: High-dimensional evaluation of the DCI algorithm in both simulation and real data; $(a) - (b)$ are the ROC curves for estimating the D-DAG $\Delta$ and its skeleton $\bar{\Delta}$ with $p = 100$ nodes, expected neighbourhood size $s = 10$, $n = 300$ samples, and 5% change between DAGs; $(c)$ shows the estimated D-DAG between gene expression data from naive and activated T cells.

estimated DAGs tend to look more similar to the true model than the incorrect estimates of GES (as also reported in [35]) and can still lead to a correct estimate of the D-DAG.

In Figure 1 (c) we analyzed the effect of changes in the noise variances on estimation performance. We set $\epsilon^{(1)} \sim \mathcal{N}(0, \mathbf{1}_p)$, while for $\epsilon^{(2)}$ we randomly picked $v$ nodes and uniformly sampled their variances from $[1.25, 2]$. We used $\alpha = .05$ as significance level based on the evaluation from Figure 1. In line with Theorem 4.4, as we increase the number of nodes $i$ such that $\epsilon_i^{(1)} \neq \epsilon_i^{(2)}$, the number of edges whose orientations can be determined decreases. This is because Algorithm 3 can only determine an edge's orientation when the variance of at least one of its nodes is invariant. Moreover, Figure 1 (c) shows that the accuracy of Algorithm 2 is not impacted by changes in the noise variances.

Finally, Figure 2 (a) - (b) show the performance (using ROC curves) of the DCI algorithm in the high-dimensional setting when initiated using KLIEP (DCI-K) and DCI-C. The simulations were performed on graphs with $p = 100$ nodes, expected neighborhood size of $s = 10$, sample size $n = 300$, and $\epsilon^{(1)}, \epsilon^{(2)} \sim \mathcal{N}(0, \mathbf{1}_p)$. $B^{(2)}$ was derived from $B^{(1)}$ so that the total number of changes was 5% of the total number of edges in $B^{(1)}$, with an equal amount of insertions and deletions. Figure 2 (a) - (b) show that both DCI-C and DCI-K perform similarly well and outperform separate estimation using GES and the PC algorithm. The respective plots for 10% change between $B^{(1)}$ and $B^{(2)}$ are given in the Supplementary Material.

## 5.2 Real data analysis

**Ovarian cancer.** We tested our method on an ovarian cancer data set [37] that contains two groups of patients with different survival rates and was previously analyzed using the DPM algorithm in the undirected setting [43]. We followed the analysis of [43] and applied the DCI algorithm to gene expression data from the apoptosis and TGF-$\beta$ pathways. In the apoptosis pathway we identified two hub nodes: BIRC3, also discovered by DPM, is an inhibitor of apoptosis [12] and one of the main disregulated genes in ovarian cancer [13]; PRKAR2B, not identified by DPM, has been shown to be important in disease progression in ovarian cancer cells [4] and an important regulatory unit for cancer cell growth [5]. In addition, the RII-$\beta$ protein encoded by PRKAR2B has been considered as a therapeutic target for cancer therapy [6, 22], thereby confirming the relevance of our findings. With respect to the TGF-$\beta$ pathway, the DCI method identified THBS2 and COMP as hub nodes. Both of these genes have been implicated in resistance to chemotherapy in epithelial ovarian cancer [19] and were also recovered by DPM. Overall, the D-UG discovered by DPM is comparable to the D-DAG found by our method. More details on this analysis are given in the Supplementary Material.

**T cell activation.** To demonstrate the relevance of our method for current genomics applications, we applied DCI to single-cell gene expression data of naive and activated T cells in order to study the pathways involved during the immune response to a pathogen. We analyzed data from 377 activated and 298 naive T cells obtained by [34] using the recent drop-seq technology. From the previously identified differentially expressed genes between naive and activated T cells [32], we selected all genes that had a fold expression change above 10, resulting in 60 genes for further analysis.

We initiated DCI using KLIEP, thresholding the edge weights at 0.005, and ran DCI for different tuning parameters and with cross-validation to obtain the final DCI output shown in Figure 2 (c) using stability selection as described in [21]. The genes with highest out-degree, and hence of interest for future interventional experiments, are GZMB and UHRF1. Interestingly, GZMB is known to induce cytotoxicity, important for attacking and killing the invading pathogens. Furthermore, this gene has been reported as the most differentially expressed gene during T cell activation [10, 26]. UHRF1 has been shown to be critical for T cell maturation and proliferation through knockout experiments [7, 24]. Interestingly, the UHRF1 protein is a transcription factor, i.e. it binds to DNA sequences and regulates the expression of other genes, thereby confirming its role as an important causal regulator. Learning a D-DAG as opposed to a D-UG is crucial for prioritizing interventional experiments. In addition, the difference UG for this application would not only have been more dense, but it would also have resulted in additional hub nodes such as FABP5, KLRC1, and ASNS, which based on the current biological literature seem secondary to T cell activation (FABP5 is involved in lipid binding, KLRC1 has a role in natural killer cells but not in T cells, and ASNS is an asparagine synthetase gene). The difference DAGs learned by separately applying the GES and PC algorithms on naive and activated T cell data sets as well as on the ovarian cancer data sets are included in the Supplementary Material for comparison.

## 6    Discussion

We presented an algorithm for directly estimating the difference between two causal DAG models given i.i.d. samples from each model. To our knowledge this is the first such algorithm and is of particular interest for learning differences between related networks, where each network might be large and complex, while the difference is sparse. We provided consistency guarantees for our algorithm and showed on synthetic and real data that it outperforms the naive approach of separately estimating two DAG models and taking their difference. While our proofs were for the setting with no latent variables, they extend to the setting where the edge weights and noise terms of all latent variables remain invariant across the two DAGs. We applied our algorithm to gene expression data in bulk and from single cells, showing that DCI is able to identify biologically relevant genes for ovarian cancer and T-cell activation. This purports DCI as a promising method for identifying intervention targets that are causal for a particular phenotype for subsequent experimental validation. A more careful analysis with respect to the D-DAGs discovered by our DCI algorithm is needed to reveal its impact for scientific discovery.

In order to make DCI scale to networks with thousands of nodes, an important challenge is to reduce the number of hypothesis tests. As mentioned in Remark 4.6, currently the time complexity (given by the number of hypothesis tests) of DCI scales exponentially with respect to the size of $S_\Theta$. The PC algorithm overcomes this problem by dynamically updating the list of CI tests given the current estimate of the graph. It is an open problem whether one can similarly reduce the number of hypothesis tests for DCI. Another challenge is to relax Assumptions 4.1 and 4.2. Furthermore, in many applications (e.g., when comparing normal to disease states), there is an imbalance of data/prior knowledge for the two models and it is of interest to develop methods that can make use of this for learning the differences between the two models.

Finally, as described in Section 2, DCI is preferable to separate estimation methods like PC and GES since it can infer not only edges that appear or disappear, but also edges with changed edge weights. However, unlike separate estimation methods, DCI relies on the assumption that the two DAGs share a topological order. Developing methods to directly estimate the difference of two DAGs that do not share a topological order is of great interest for future work.

## Acknowledgements

Yuhao Wang was supported by ONR (N00014-17-1-2147), NSF (DMS-1651995) and the MIT-IBM Watson AI Lab. Anastasiya Belyaeva was supported by an NSF Graduate Research Fellowship (1122374) and the Abdul Latif Jameel World Water and Food Security Lab (J-WAFS) at MIT. Caroline Uhler was partially supported by ONR (N00014-17-1-2147), NSF (DMS-1651995), and a Sloan Fellowship.

## Footnotes

[1] This is similar to the faithfulness assumption in the Gaussian setting, where partial correlations are used for CI testing; the partial correlations are rational functions in the variables $B_{ij}^{(k)}$ and $\sigma_j^{(k)}$ and the faithfulness assumption asserts that if a partial correlation $\rho_{ij|S}$ is zero then the corresponding rational function is identically equal to zero and hence $B_{ij} = 0$ [16].

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
