[Supplementary Material]

# Supplementary Material: Direct Estimation of Differences in Causal Graphs

**Yuhao Wang**
Lab for Information & Decision Systems
and Institute for Data, Systems and Society
Massachusetts Institute of Technology
Cambridge, MA 02139
yuhaow@mit.edu

**Chandler Squires**
Lab for Information & Decision Systems
and Institute for Data, Systems and Society
Massachusetts Institute of Technology
Cambridge, MA 02139
csquires@mit.edu

**Anastasiya Belyaeva**
Lab for Information & Decision Systems
and Institute for Data, Systems and Society
Massachusetts Institute of Technology
Cambridge, MA 02139
belyaeva@mit.edu

**Caroline Uhler**
Lab for Information & Decision Systems
and Institute for Data, Systems and Society
Massachusetts Institute of Technology
Cambridge, MA 02139
cuhler@mit.edu

## A   Hypothesis testing framework

In this section, we provide the details regarding the hypothesis tests that we used for testing the following two null hypotheses:

$$H_0^{i,j|S} : \beta_{i,j|S}^{(1)} = \beta_{i,j|S}^{(2)} \qquad \text{and} \qquad H_0^{j|S} : \sigma_{j|S}^{(1)} = \sigma_{j|S}^{(2)}.$$

As in [6], we used hypothesis tests based on the F-test for testing for invariance between regression coefficients and residual variances. For testing $H_0^{i,j|S} : \beta_{i,j|S}^{(1)} = \beta_{i,j|S}^{(2)}$ we used the test statistic

$$\hat{T} := (\hat{\beta}_{i,j|S}^{(1)} - \hat{\beta}_{i,j|S}^{(2)})^2 \cdot \quad \left[ \left( (\hat{\sigma}_{j|M}^{(1)})^2 (n_1 \hat{\Sigma}_{M,M}^{(1)})^{-1} + (\hat{\sigma}_{j|M}^{(2)})^2 (n_2 \hat{\Sigma}_{M,M}^{(2)})^{-1} \right)^{-1} \right]_{i_M i_M}$$

where $\hat{\beta}_{i,j|S}^{(k)}$ is the empirical estimate of $\beta_{i,j|S}^{(k)}$ obtained by ordinary least squares, $(\hat{\sigma}_{j|M}^{(k)})^2$ is an unbiased estimator of the regression residual variance $(\sigma_{j|M}^{(k)})^2$, $\hat{\Sigma}_{M,M}^{(k)}$ is the sample covariance matrix of the random vector $X_M^{(k)}$ with $M = \{i\} \cup S$, and $i_M$ denotes the index in $M$ corresponding to the element $i$. In [11][Section 3.6], the author shows that under the null hypothesis the asymptotic distribution of $\hat{T}$ can be approximated by the F-distribution $F(1, n_1 + n_2 - 2|S| - 2)$. The basic explanation is that, for $M := S \cup \{i\}$, let $\beta_M^{(k)}$ be the best linear predictor when regressing $X_j^{(k)}$ onto $X_M^{(k)}$, i.e., our estimator is $X_j^{(k)} = (\beta_M^{(k)})^T X_M^{(k)} + \tilde{\epsilon}_j^{(k)}$. Let $\beta$ be the vector

$$\beta := \begin{bmatrix} \beta_M^{(1)} \\ \beta_M^{(2)} \end{bmatrix},$$

and let $C \in \mathbb{R}^{2|M|}$ have $C_{i_M} = 1$ and $C_{|M|+i_M} = -1$ and all other entries as zero. Then the null hypothesis $H_0^{i,j|S}$ can be written as: $C^T \beta = 0$. It follows from Proposition 3.5 of [11], on the asymptotic distribution of the Wald statistic, that $\hat{T}$ converges in distribution to $\chi^2(1)$, i.e., a $\chi^2$-distribution with 1 degree of freedom.

However, the F-distribution $F(1, n_1 + n_2 - 2|S| - 2)$ is a better approximation for the distribution of $\hat{T}$, as outlined in Section 3.6 of [11]. A brief justification is in order. First, we know that the

convergence is the same: for an F-distribution $F(1, d)$, as $d \to \infty$, we have $F(1, d) \overset{d}{\to} \chi^2(1)$. Additionally, $F(1, d)$ and $\hat{T}$ both have a fatter tail than $\chi^2(1)$. Together, these facts suggest the choice of a F-distribution $F(1, d)$ with $d \to \infty$ as $n_1, n_2 \to \infty$. For the second parameter $d$, we used $d = n_1 + n_2 - 2|S| - 2$, the total degrees of freedom of the unbiased estimators of the two regression residual variances, i.e., $(\hat{\sigma}_{j|M}^{(1)})^2$ and $(\hat{\sigma}_{j|M}^{(2)})^2$.

Similarly, for testing $H_0^{j|S}$, we used the test statistic

$$\hat{F} := (\hat{\sigma}_{j|S}^{(1)})^2 / (\hat{\sigma}_{j|S}^{(2)})^2.$$

Under the null hypothesis, $\hat{F}$ is a ratio of two $\chi^2$-distributed random variables and hence $\hat{F}$ follows an F-distribution, namely $F(n_1 - |S| - 1, n_2 - |S| - 1)$.

## B  Comparison to related work on invariant causal structure learning

The complimentary problem to learning the difference of two DAG models is the problem of inferring the causal structure that is *invariant* across different environments. Algorithms for this problem have been developed in recent literature [6, 16, 20]. Since the hypothesis testing framework in [6] is similar to our approach, we here provide an example to explain the differences between the two approaches and in particular to show that a new approach is needed in order to obtain a consistent method for learning the difference DAG.

Recall that when we have access to data from a pair of DAGs, the algorithms in [6] make use of the following two sets that are estimated from the data. The first is the *regression invariance set*:

$$R := \left\{ (j, S) : \beta_S^{(1)}(j) = \beta_S^{(2)}(j) \right\},$$

where $\beta_S^{(k)}(j)$ corresponds to the best linear predictor when regressing $X_j^{(k)}$ onto $X_S^{(k)}$. The second is $I$, the set of variables whose internal noise variances have been changed across the two DAGs:

$$I := \left\{ j : \forall S \subseteq [p] \setminus \{j\}, \mathbb{E}(X_j^{(1)} - (\beta_S^{(1)}(j))^T X_S^{(1)})^2 \neq \mathbb{E}(X_j^{(2)} - (\beta_S^{(2)}(j))^T X_S^{(2)})^2 \right\}.$$

The output of the algorithms in [6] is fully determined by the invariant elements given in $R$ and $I$. In particular, Algorithm 1 in [6] estimates the invariant causal structure by considering all elements in $R$ and $I$, while Algorithm 2 in [6] is a more efficient constraint-based algorithm that considers only a subset of the elements in $R$.

(a)  (b)

(c)

Figure B.1: (a) - (b): Example of two DAG pairs where the corresponding D-DAGs are different but the application of algorithms 1 and 2 from [6] would result in the same sets $R$ and $I$. The red edges correspond to the edges that have different edge weights across the two DAGs, the black edges correspond to the edges that have the same edge weights across the two DAGs. The red nodes correspond to the nodes that have different internal noise variances across the two DAGs and the black nodes correspond to the nodes that have unchanged internal noise variances. (c): D-DAGs output by the DCI algorithm when data is generated from (a) and (b), respectively.

**Example B.1.** *Figure B.1 shows two cases where the underlying D-DAGs are different but in both cases [6] would produce the same sets $R$ and $I$ that are used to assign edge orientations. In (a) we consider two fully connected linear SEMs $(B^{(1)}, \epsilon^{(1)})$ and $(B^{(2)}, \epsilon^{(2)})$ where the edge weights of all edges change across the two DAGs. The variances of the internal noise terms for nodes $1$ and $3$ change while the variance of the internal noise term of node $2$ stays the same. In (b) we instead consider two fully connected linear SEMs $(B^{(1)}, \epsilon^{(1)})$ and $(B^{(2)}, \epsilon^{(2)})$ where $B_{12}^{(1)} \neq B_{12}^{(2)}$ and $B_{23}^{(1)} \neq B_{23}^{(2)}$. Moreover, the variances of the internal noise terms of nodes $1$ and $3$ change across $k = \{1, 2\}$ while the variance of node $2$ stays the same. It can be easily shown that in both cases $R = \emptyset$ and $I = \{1, 3\}$. Since both (a) and (b) correspond to exactly the same $R$ and $I$, by simply using the output from [6], we cannot distinguish whether the data is generated from the pair of DAGs given in (a) or the pair of DAGs given in (b). In fact, since for these examples $R$ is empty, [6] will not uncover any edge orientations consistent with the underlying DAGs $\mathcal{G}^{(1)}$ or $\mathcal{G}^{(2)}$. On the other hand, our algorithm is able to distinguish these two cases as well as discover the edge orientations of the underlying D-DAGs, as shown in Figure B.1 (c).*

## C  Theoretical analysis

### C.1  Preliminaries: Schur complement

In this section, we describe how to use Schur complements to express $\beta_{i,j|S}^{(k)}$ and $(\sigma_{j|S}^{(k)})^2$ as rational functions in the variables $(B_{ij}^{(k)})_{(i,j) \in A^{(k)}}$ and $(\sigma_j^{(k)})_{j \in [p]}$. This will be used for the proofs of Theorems 4.3 and 4.4 in the following sections.

For a subset of nodes $M \subseteq [p]$, let $X_M$ denote the random vector spanned by the random variables $X_i$ for all $i \in M$. Let $\neg M$ denote the complement of $M$ with respect to the full set of nodes, i.e., $\neg M := [p] \setminus M$. The inverse covariance matrix of the random vector $X_M$, i.e., $(\Sigma_{M,M})^{-1}$, can be obtained from $\Theta$ by taking the Schur complement:

$$\Theta_M := (\Sigma_{M,M})^{-1}$$
$$= \Theta_{M,M} - \Theta_{M,\neg M}(\Theta_{\neg M, \neg M})^{-1}\Theta_{\neg M, M}.$$

Note that here $\Theta_M$ does not represent the submatrix of $\Theta$ with set of row and column indices in $M$, i.e., $\Theta_{M,M}$, but rather the Schur complement. For any two indices $i, j \in M$, let $i_M, j_M \in [|M|]$ denote the row/column indices of matrix $\Theta_M$ associated to $i$ and $j$, then the $(i_M, j_M)$-th entry of matrix $\Theta_M$ can be written as:

$$(\Theta_M)_{i_M j_M} = \Theta_{ij} - \Theta_{i, \neg M}(\Theta_{\neg M, \neg M})^{-1}\Theta_{\neg M, j}.$$

In [4, 10, 19] the authors also give a combinatorial characterization of the Schur complement. Following their characterization, the value of $(\Theta_M)_{i_M j_M}$ is determined by the parameters of the *d-connecting paths* from node $i$ to $j$ given $M \setminus \{i, j\}$. In this case, the entry $(\Theta_M^{(k)})_{i_M j_M}$ would be invariant for $k = \{1, 2\}$ if the parameters along the d-connecting paths are all the same. Finally, by applying the result of [17], $\beta_{i,j|S}^{(k)}$ and $(\sigma_{j|S}^{(k)})^2$ can be written as:

$$\beta_{i,j|S}^{(k)} = -\frac{(\Theta_M^{(k)})_{i_M j_M}}{(\Theta_M^{(k)})_{j_M j_M}} \quad \text{where} \quad M = S \cup \{i, j\},$$

$$(\sigma_{j|S}^{(k)})^2 = \left((\Theta_M^{(k)})_{j_M j_M}\right)^{-1} \quad \text{where} \quad M = S \cup \{j\}. \tag{S.1}$$

Combining Eq. (S.1) with the formula for the Schur complement, one can easily see that $\beta_{i,j|S}^{(k)}$ and $(\sigma_{j|S}^{(k)})^2$ can be expressed as rational functions in the variables $(B_{ij}^{(k)})_{(i,j) \in A^{(k)}}$ and $(\sigma_j^{(k)})_{j \in [p]}$.

### C.2  Proof of Theorem 4.3

In this Section we provide the consistency proofs of Theorem 4.3 when Algorithm 2 is initialized in the difference-UG. The proof of Theorem 4.3 when Algorithm 2 is initialized in the complete graph follows easily from the proofs in this section. To complete the proof, one also needs the following assumption:

**Assumption C.1** (Difference-precision-matrix-faithfulness assumption). *For any choices of $i, j \in [p]$, it holds that*

1. *If $B_{ij}^{(1)} \neq B_{ij}^{(2)}$, then $\Theta_{ij}^{(1)} \neq \Theta_{ij}^{(2)}$, and for any $\ell$ with directed path $i \rightarrow j \leftarrow \ell$ in either $\mathcal{G}^{(1)}$ or $\mathcal{G}^{(2)}$, $\ell \in S_\Theta$.*

2. *If $\sigma_j^{(1)} \neq \sigma_j^{(2)}$, then $\Theta_{jj}^{(1)} \neq \Theta_{jj}^{(2)}$, and $\forall\, i \in \mathrm{Pa}^{(1)}(j) \cup \mathrm{Pa}^{(2)}(j), i \in S_\Theta$.*

Note that Assumption C.1 is not a necessary assumption for the consistency of Algorithm 2, since one can simply take $\Delta_\Theta$ as the fully connected graph on $p$ nodes and $S_\Theta = [p]$ as input. The same holds for the proof of Theorem 4.4. The strength of Assumption C.1 is further analyzed in Remark C.5.

To prove Theorem 4.3, we need to make use of the following two lemmas:

**Lemma C.2.** *Given $\Theta^{(1)}$ and $\Theta^{(2)}$, if $\Theta_{ij}^{(1)} = \Theta_{ij}^{(2)}$, then $(\Theta_M^{(1)})_{i_M j_M} = (\Theta_M^{(2)})_{i_M j_M}$ for $M = S_\Theta \cup \{i, j\}$.*

*Proof.* By Schur complement, we have that $(\Theta_M^{(k)})_{i_M j_M} = \Theta_{ij}^{(k)} - \Theta_{i,\neg M}^{(k)}(\Theta_{\neg M,\neg M}^{(k)})^{-1}\Theta_{\neg M,j}^{(k)}$. By the definition of $S_\Theta$, $\Theta_{M,\neg M}^{(1)} = \Theta_{M,\neg M}^{(2)}$ and $\Theta_{\neg M,\neg M}^{(1)} = \Theta_{\neg M,\neg M}^{(2)}$. $\qquad\square$

**Lemma C.3.** *Given two linear SEMs $(B^{(1)}, \epsilon^{(1)})$ and $(B^{(2)}, \epsilon^{(2)})$ and denoting the precision preci-sion matrix of the random vector $X_{1:j}^{(k)}$ by $\Theta^{*(k)}$, then under Assumption C.1 we have $S_{\Theta^*} \subseteq S_\Theta$.*

*Proof.* Since $B^{(k)}$ is strictly upper triangular, the marginal distribution of the random vector $X_{1:j}^{(k)}$ follows a new SEM,

$$X_{1:j}^{(k)} = (B_{1:j,1:j}^{(k)})^T X_{1:j}^{(k)} + \epsilon_{1:j}^{(k)},$$

where $B_{1:j,1:j}^{(k)}$ is the submatrix of $B^{(k)}$ with the first $j$ rows and $j$ columns, and $\epsilon_{1:j}^{(k)}$ is the random vector with the first $j$ random variables of $\epsilon^{(k)}$. It can then be shown that the $(i, \ell)$-th entry of the new precision matrix $\Theta^*$ is given by:

$$\Theta_{i\ell}^{*(k)} = -(\sigma_\ell^{(k)})^{-2} B_{i\ell}^{(k)} + \sum_{\ell < m \leq j} (\sigma_m^{(k)})^{-2} B_{im}^{(k)} B_{\ell m}^{(k)}.$$

It is then a short exercise to show that $\Theta_{i\ell}^{*(1)} \neq \Theta_{i\ell}^{*(2)}$ only if at least one of the following two statements hold:

1. $B_{i\ell}^{(1)} \neq B_{i\ell}^{(2)}$ or $\sigma_\ell^{(1)} \neq \sigma_\ell^{(2)}$;

2. There exists at least one of $\ell < m \leq j$ with $i \rightarrow m \leftarrow \ell$ in either $\mathcal{G}^{(1)}$ or $\mathcal{G}^{(2)}$ such that $B_{im}^{(1)} \neq B_{im}^{(2)}$ or $B_{\ell m}^{(1)} \neq B_{\ell m}^{(2)}$ or $\sigma_m^{(1)} \neq \sigma_m^{(2)}$.

By applying Assumption C.1, we have that $\Theta_{i\ell}^{*(1)} \neq \Theta_{i\ell}^{*(2)} \Rightarrow i, \ell \in S_\Theta$.

The diagonal entries of the precision matrix are given by:

$$\Theta_{ii}^{*(k)} = (\sigma_i^{(k)})^{-2} + \sum_{i < m \leq j} (\sigma_m^{(k)})^{-2} B_{im}^{(k)}.$$

Clearly, $\Theta_{ii}^{*(1)} \neq \Theta_{ii}^{*(2)}$ only if at least one of the following statements hold:

1. $\sigma_i^{(1)} \neq \sigma_i^{(2)}$;

2. $B_{im}^{(1)} \neq B_{im}^{(2)}$ or $\sigma_m^{(1)} \neq \sigma_m^{(2)}$ for at least one of the descendents of $i$ in either $\mathcal{G}^{(1)}$ or $\mathcal{G}^{(2)}$ with $i < m \leq j$.

By applying Assumption C.1 we have that $\Theta_{ii}^{*(1)} \neq \Theta_{ii}^{*(2)} \Rightarrow i \in S_\Theta$. $\qquad\square$

**Lemma C.4.** *Given two linear SEMs $(B^{(1)}, \epsilon^{(1)})$ and $(B^{(2)}, \epsilon^{(2)})$, then under Assumption 4.1, $B_{ij}^{(1)} = B_{ij}^{(2)}$ if and only if*

$$\exists S \subseteq S_\Theta \setminus \{i, j\} \text{ s.t. } \beta_{i,j|S}^{(1)} = \beta_{i,j|S}^{(2)} \text{ or } \beta_{j,i|S}^{(1)} = \beta_{j,i|S}^{(2)}.$$

*Proof.* We show the "if" direction by proving the contrapositive, i.e. if $B_{ij}^{(1)} \neq B_{ij}^{(2)}$, then

$$\forall S \subseteq S_\Theta \setminus \{i, j\}, \ \beta_{i,j|S}^{(1)} \neq \beta_{i,j|S}^{(2)} \text{ and } \beta_{j,i|S}^{(1)} \neq \beta_{j,i|S}^{(2)}. \tag{S.2}$$

This follows directly from Assumption 4.1.

Now, we prove the "only if" direction, i.e., if $B_{ij}^{(1)} = B_{ij}^{(2)}$, then

$$\exists S \subseteq S_\Theta \setminus \{i, j\} \text{ s.t. } \beta_{i,j|S}^{(1)} = \beta_{i,j|S}^{(2)} \text{ or } \beta_{j,i|S}^{(1)} = \beta_{j,i|S}^{(2)}.$$

We divide the proof into two cases: $\sigma_j^{(1)} = \sigma_j^{(2)}$, and $\sigma_j^{(1)} \neq \sigma_j^{(2)}$.

**Case 1** $\sigma_j^{(1)} = \sigma_j^{(2)}$

Consider the precision matrix $\Theta^{*(k)}$ of the random vector $X_{1:j}^{(k)}$. In this case, we prove that choosing the conditioning set $S = S_{\Theta^*} \setminus \{i, j\}$ implies regression invariance. This is a valid choice for $S$, since it is a subset of $S_\Theta \setminus \{i, j\}$ by Lemma C.3.

We will first show that $\Theta_{ij}^{*(1)} = \Theta_{ij}^{*(2)}$ and $\Theta_{jj}^{*(1)} = \Theta_{jj}^{*(2)}$. According to the new SEM of the marginal distribution of the random vector $X_{1:j}^{(k)}$, i.e.,

$$X_{1:j}^{(k)} = (B_{1:j,1:j}^{(k)})^T X_{1:j}^{(k)} + \epsilon_{1:j}^{(k)},$$

it is easy to conclude that node $j$ no longer has any descendants in the marginal SEM. We therefore have that

$$\Theta_{ij}^{*(k)} = -(\sigma_j^{(k)})^{-2} B_{ij}^{(k)} \quad \text{and} \quad \Theta_{jj}^{*(k)} = (\sigma_j^{(k)})^{-2}.$$

Since $B_{ij}^{(1)} = B_{ij}^{(2)}$ and $\sigma_j^{(1)} = \sigma_j^{(2)}$, then

$$\Theta_{ij}^{*(1)} = \Theta_{ij}^{*(2)} \quad \text{and} \quad \Theta_{jj}^{*(1)} = \Theta_{jj}^{*(2)}. \tag{S.3}$$

By choosing $M := S \cup \{i, j\}$ and denoting $M^* := [j] \setminus M$, recall that the entries of $\Theta_M^{(k)}$ can be written as

$$(\Theta_M^{(k)})_{i_M j_M} = \Theta_{ij}^{*(k)} - \Theta_{i,M^*}^{*(k)} (\Theta_{M^*,M^*}^{*(k)})^{-1} \Theta_{M^*,j}^{*(k)}.$$

Now by invoking Lemma C.2 and Eq. (S.3), we obtain that $(\Theta_M^{(1)})_{i_M j_M} = (\Theta_M^{(2)})_{i_M j_M}$ and $(\Theta_M^{(1)})_{j_M j_M} = (\Theta_M^{(2)})_{j_M j_M}$. Finally, using Eq. (S.1), we obtain $\beta_{i,j|S}^{(1)} = \beta_{i,j|S}^{(2)}$.

**Case 2** $\sigma_j^{(1)} \neq \sigma_j^{(2)}$

In this case, we prove that regressing on all of the parents of $j$ in both DAGs, i.e., choosing the conditioning set as $S = \mathrm{Pa}^{(1)}(j) \cup \mathrm{Pa}^{(2)}(j) \setminus \{i\}$, implies regression invariance. This is a valid choice for $S$, i.e. $S \subseteq S_\Theta \setminus \{i, j\}$, since Assumption C.1 ensures that if $\sigma_j^{(1)} \neq \sigma_j^{(2)}$ then $\ell \in S_\Theta$ for all $\ell \in \mathrm{Pa}^{(k)}(j)$.

Let $M := S \cup \{i\}$. By regressing $X_j^{(k)}$ onto $X_M^{(k)}$, we get the regression coefficient as

$$X_j^{(k)} = (\beta_M^{(k)})^T X_M^{(k)} + \tilde{\epsilon}_j^{(k)}.$$

Let $(\beta_M^{(k)})_{\ell_M}$ denote the $\ell_M$-th entry of $\beta_M^{(k)}$. By the Markov property, when regressing $X_j^{(k)}$ onto $X_M^{(k)}$ where $\mathrm{Pa}^{(k)}(j) \subseteq M \subseteq [j-1]$, it is guaranteed that $(\beta_M^{(k)})_{\ell_M} = B_{\ell j}^{(k)}$ if $\ell \in \mathrm{Pa}^{(k)}(j)$ and $(\beta_M^{(k)})_{\ell_M} = 0$ otherwise. Therefore, we have that $\beta_{i,j|S}^{(k)} = (\beta_M^{(k)})_{i_M} = B_{ij}^{(k)}$, which completes the proof. $\square$

We now show how the proof of Theorem 4.3 follows from this lemma.

*Proof.* By applying Assumption C.1 we have that $\bar{\Delta} \subseteq \Delta_\Theta$. Then the proof of Theorem 4.3 follows trivially from Lemma C.4, since Lemma C.4 shows that an edge $i - j$ is deleted during testing the invariance of regression coefficients if and only if $i - j \notin \bar{\Delta}$. $\qquad\square$

We end this section with a remark about the strength of Assumption C.1.

**Remark C.5** (Strength of Assumption C.1). *To analyze the strength of Assumption C.1, consider instead the following stronger assumption:*

**Assumption C.1'** *For any choices of $i, j \in [p]$, it holds that*

1. *If $B_{ij}^{(1)} \neq B_{ij}^{(2)}$, then $\Theta_{ij}^{(1)} \neq \Theta_{ij}^{(2)}$, and $\Theta_{i\ell}^{(1)} \neq \Theta_{i\ell}^{(2)}$ for any $\ell$ with directed path $i \rightarrow j \leftarrow \ell$ in either $\mathcal{G}^{(1)}$ or $\mathcal{G}^{(2)}$.*

2. *If $\sigma_j^{(1)} \neq \sigma_j^{(2)}$, then $\Theta_{jj}^{(1)} \neq \Theta_{jj}^{(2)}$, and $\Theta_{ii}^{(1)} \neq \Theta_{ii}^{(2)} \, \forall \, i \in \mathrm{Pa}^{(1)}(j) \cup \mathrm{Pa}^{(2)}(j)$.*

*Assumption C.1' is a strictly stronger assumption than Assumption C.1, i.e., Assumption C.1 is satisfied whenever Assumption C.1' is satisfied. We expect Assumption C.1' to be much weaker than Assumptions 4.1 and 4.2 in the finite sample regime, and therefore the same also holds for Assumption C.1. This is because the number of hypersurfaces violating Assumption C.1' scales at most as $\mathcal{O}(p^4)$, which is a much smaller number as compared to Assumptions 4.1 and 4.2 that scale as $\mathcal{O}(|\Delta_\Theta| 2^{|S_\Theta|-1})$.* $\qquad\square$

## C.3 Proof of Theorem 4.4

In this section, we provide a proof of Theorem 4.4 when Algorithm 3 is initialized in the difference-UG.

**Lemma C.6.** *For all nodes $j$ incident to at least one edge in $\bar{\Delta}$, $\sigma_j^{(1)} = \sigma_j^{(2)}$ if and only if $\exists \, S \subseteq S_\Theta \setminus \{i, j\}$ s.t. $\sigma_{j|S}^{(1)} = \sigma_{j|S}^{(2)}$.*

*Proof.* Proving the "if" direction is equivalent to showing that, if $\sigma_j^{(1)} \neq \sigma_j^{(2)}$, then

$$\forall S \subseteq S_\Theta \setminus \{j\}, \; \sigma_{j|S}^{(1)} \neq \sigma_{j|S}^{(2)}. \tag{S.4}$$

This follows directly from Assumption 4.2.

To prove the "only if" direction, consider again the marginal distribution of $X_{1:j}^{(k)}$. Since $\sigma_j^{(1)} = \sigma_j^{(2)}$, we have that $\Theta_{jj}^{*(1)} = \Theta_{jj}^{*(2)}$. Let $M := S_{\Theta^*} \cup \{j\}$ and let $S := M \setminus \{j\}$, since $(\sigma_{j|S}^{(k)})^2 = ((\Theta_M^{(k)})_{j_M j_M})^{-1}$ and $(\Theta_M^{(1)})_{j_M j_M} = (\Theta_M^{(2)})_{j_M j_M}$ by using Lemma C.2, we have that $\sigma_{j|S}^{(1)} = \sigma_{j|S}^{(2)}$. $\qquad\square$

**Lemma C.7.** $\forall \, i - j \in \bar{\Delta}$ *such that $\sigma_j^{(1)} = \sigma_j^{(2)}$ it holds that,*

1. *if $i \rightarrow j \in \Delta$, then $i \in S$ for all $S$ s.t. $\sigma_{j|S}^{(1)} = \sigma_{j|S}^{(2)}$.*

2. *if $j \rightarrow i \in \Delta$, then $i \notin S$ for all $S$ s.t. $\sigma_{j|S}^{(1)} = \sigma_{j|S}^{(2)}$.*

*Proof.* We prove both statements by contradiction. For $B_{ij}^{(1)} \neq B_{ij}^{(2)}$, suppose there exists a $S$ such that $\sigma_{j|S}^{(1)} = \sigma_{j|S}^{(2)}$ while $i \notin S$. This contradicts Assumption 4.2.

Similarly, in the second statement for $B_{ji}^{(1)} \neq B_{ji}^{(2)}$, suppose there exists $S$ such that $\sigma_{j|S \cup \{i\}}^{(1)} = \sigma_{j|S \cup \{i\}}^{(2)}$. This also contradicts Assumption 4.2. $\qquad\square$

We now show how the proof of Theorem 4.4 follows from this lemma.

$$\mathcal{G}^{(1)} \qquad\qquad \mathcal{G}^{(2)}$$

$$1 \xrightarrow{0.5} 2 \qquad\qquad 1 \xrightarrow{0.5} 2$$
$$\text{-0.25} \qquad 0.5 \qquad\qquad 0.1 \qquad 0.5$$
$$3 \qquad\qquad\qquad\qquad 3$$

Figure D.1: Example of two linear SEMs that satisfy Assumpitons 4.1 and 4.2 but do not satisfy the faithfulness assumption. The autoregressive matrices $B^{(1)}$ and $B^{(2)}$ are shown as edge weights in $\mathcal{G}^{(1)}$ and $\mathcal{G}^{(2)}$. We assume that all noise terms are standard normal random variables.

*Proof.* By Lemma C.6, there exists $S$ such that $\sigma_{j|S}^{(1)} = \sigma_{j|S}^{(2)}$ if and only if $\sigma_j^{(1)} = \sigma_j^{(2)}$. Therefore, all the nodes where the internal noise variance is unchanged will be chosen by Algorithm 3. In addition, it also follows from Lemma C.7 that for any $i \to j \in \Delta$, $i \in S$ and for any $j \to i \in \Delta$, $i \notin S$. This proves that for any node $i$ where $\sigma_i^{(k)}$ is invariant, all edges adjacent to $i$ are oriented and that all edges oriented before the last step of Algorithm 3 are correctly oriented.

It remains to show that all edges oriented in the last step of Algorithm 3 are correct. This easily follows from the acyclicity property of the underlying graphs and from the fact that all edge orientations before the last step are correct. $\qquad\square$

## D  Examples for Remark 4.7

Since our assumptions are closely related to the faithfulness assumption, it is interesting to compare the entailment relationship between our assumptions, i.e., Assumptions 4.1 and 4.2, and the faithfulness assumption. In this section, we give the following two counterexamples to show that our assumptions and the faithfulness assumption do not imply one another.

**Example D.1.** *We give a 3-node example that satisfies Assumptions 4.1 and 4.2 but does not satisfy the faithfulness assumption. Consider two linear SEMs $(B^{(1)}, \epsilon^{(1)})$ and $(B^{(2)}, \epsilon^{(2)})$ with $\epsilon_j^{(k)} \sim \mathcal{N}(0,1)\ \forall\ j,k$ and where $B^{(1)}$ and $B^{(2)}$ are the autoregressive matrices defined as shown in Figure D.1. Clearly, $\mathbb{P}^{(1)}$ does not satisfy the faithfulness assumption with respect to $\mathcal{G}^{(1)}$ since nodes 1 and 3 are d-connected given $\emptyset$, but $X_1^{(1)} \perp\!\!\!\perp X_3^{(1)}$. However, it is a short exercise to show that for all choices of $S$, i.e. $\emptyset$ and $\{2\}$, we have $\beta_{1,3|S}^{(1)} \neq \beta_{1,3|S}^{(2)}$, $\beta_{3,1|S}^{(1)} \neq \beta_{3,1|S}^{(2)}$, $\sigma_{3|S}^{(1)} \neq \sigma_{3|S}^{(2)}$ and $\sigma_{1|S\cup\{3\}}^{(1)} \neq \sigma_{1|S\cup\{3\}}^{(2)}$. Therefore, this example satisfies Assumptions 4.1 and 4.2.*

**Example D.2.** *We give a 3-node example that satisfies the faithfulness assumption but does not satisfy Assumption 4.1. Consider two linear SEMs where all $\epsilon_j^{(k)}$ are standard normal random variables and $B^{(1)}$ and $B^{(2)}$ are defined as shown in Figure D.2. Although $B_{13}^{(1)} \neq B_{13}^{(2)}$, by choosing $S = \emptyset$, we still have that $\beta_{1,3|S}^{(1)} = \beta_{1,3|S}^{(2)} = 0.5$. Therefore, although both SEMs satisfy the faithfulness assumption, the pair does not satisfy Assumption 4.1.*

Next, we give an example explaining the hypersurfaces that correspond to the set of parameters violating our assumptions versus the faithfulness assumption. This example shows that the number of hypersurfaces corresponding to violations of the faithfulness assumption is much higher than the number of hypersurfaces corresponding to violations of our assumptions, which implies that the faithfulness assumption is more restrictive in the finite sample regime.

$$\mathcal{G}^{(1)} \qquad\qquad \mathcal{G}^{(2)}$$

$$1 \xrightarrow{0.5} 2 \qquad\qquad 1 \xrightarrow{0.5} 2$$
$$0.25 \qquad 0.5 \qquad\qquad 0.1 \qquad 0.8$$
$$3 \qquad\qquad\qquad\qquad 3$$

Figure D.2: Example of two linear SEMs that satisfy the faithfulness assumption but do not satisfy Assumption 4.1. The autoregressive matrices $B^{(1)}$ and $B^{(2)}$ are shown as edge weights in $\mathcal{G}^{(1)}$ and $\mathcal{G}^{(2)}$. We assume that all noise terms are standard normal random variables.

Figure D.3: Example of two fully connected linear SEMs. The red edges correspond to the edges that have different edge weights across the two DAGs, the black edges correspond to the edges that have the same edge weights across the two DAGs. The variances of internal noise terms remain the same for both DAGs.

**Example D.3.** *We give a 3-node example to provide intuition for why the number of hypersurfaces violating the faithfulness assumption is usually much higher than the number of hypersurfaces violating our assumptions. Consider the two fully connected linear SEMs $(B^{(1)}, \epsilon^{(1)})$ and $(B^{(2)}, \epsilon^{(2)})$ shown in Figure D.3. In this example, $B_{12}^{(1)} \neq B_{12}^{(2)}$ while the noise variances and all other edge weights are not changed across the two DAGs.*

*If we think of each parameter $B_{ij}^{(k)}$ or $\sigma_j^{(k)}$ not as a parameter but rather as an indeterminate, the set of parameters that violate the faithfulness assumption and our assumptions correspond to a system of polynomial equations in the following 7 indeterminates: $(B_{12}^{(1)}, B_{12}^{(2)}, B_{13}, B_{23}, \sigma_1, \sigma_2, \sigma_3)$. Note that here we use a single indeterminate $B_{13}$ to encode both the parameters $B_{13}^{(1)}$ and $B_{13}^{(2)}$ since they have the same value. The set of parameters that violate the faithfulness assumption are given by the following 11 polynomial equations and hence correspond to a collection of 11 hypersurfaces:*

$$\text{cov}(X_1^{(1)}, X_2^{(1)}): \quad B_{12}^{(1)}\sigma_1^2 = 0,$$

$$\text{cov}(X_1^{(1)}, X_3^{(1)}): \quad B_{13}\sigma_1^2 + B_{12}^{(1)}B_{23}\sigma_1^2 = 0,$$

$$\text{cov}(X_2^{(1)}, X_3^{(1)}): \quad (B_{12}^{(1)})^2 B_{23}\sigma_1^2 + B_{12}^{(1)}B_{13}\sigma_1^2 + B_{23}\sigma_2^2 = 0,$$

$$\text{cov}(X_1^{(1)}, X_2^{(1)} \mid X_3^{(1)}): \quad -\frac{B_{13}B_{23}\sigma_1^2\sigma_2^2 - B_{12}^{(1)}\sigma_1^2\sigma_3^2}{(B_{13} + B_{12}^{(1)}B_{23})^2\sigma_1^2 + B_{23}^2\sigma_2^2 + \sigma_3^2} = 0,$$

$$\text{cov}(X_1^{(1)}, X_3^{(1)} \mid X_2^{(1)}): \quad \frac{B_{13}\sigma_1^2\sigma_2^2}{(B_{12}^{(1)})^2\sigma_1^2 + \sigma_2^2} = 0,$$

$$\text{cov}(X_2^{(1)}, X_3^{(1)} \mid X_1^{(1)}): \quad B_{23}\sigma_2^2 = 0,$$

$$\text{cov}(X_1^{(2)}, X_2^{(2)}): \quad B_{12}^{(2)}\sigma_1^2 = 0,$$

$$\text{cov}(X_1^{(2)}, X_3^{(2)}): \quad B_{13}\sigma_1^2 + B_{12}^{(2)}B_{23}\sigma_1^2 = 0,$$

$$\text{cov}(X_2^{(2)}, X_3^{(2)}): \quad (B_{12}^{(2)})^2 B_{23}\sigma_1^2 + B_{12}^{(2)}B_{13}\sigma_1^2 + B_{23}\sigma_2^2 = 0,$$

$$\text{cov}(X_1^{(2)}, X_2^{(2)} \mid X_3^{(2)}): \quad -\frac{B_{13}B_{23}\sigma_1^2\sigma_2^2 - B_{12}^{(2)}\sigma_1^2\sigma_3^2}{(B_{13} + B_{12}^{(2)}B_{23})^2\sigma_1^2 + B_{23}^2\sigma_2^2 + \sigma_3^2} = 0,$$

$$\text{cov}(X_1^{(2)}, X_3^{(2)} \mid X_2^{(2)}): \quad \frac{B_{13}\sigma_1^2\sigma_2^2}{(B_{12}^{(2)})^2\sigma_1^2 + \sigma_2^2} = 0.$$

Figure D.4: Parameter values corresponding to unfaithful distributions in Example D.3; the first three figures are the hypersurfaces corresponding to $\text{cov}(X_1, X_2) = 0$, $\text{cov}(X_1, X_2 \mid X_3) = 0$ and $\text{cov}(X_1, X_3) = 0$ respectively when setting $\sigma_i = 1$ for visualization in 3d; the last figure shows the hypersurfaces of the first 6 polynomials with $\sigma_i = 1$. Figure adopted from [19, Figure 2])

*To get a better sense of how the hypersurfaces of these polynomials are distributed in the parameter space, Figure D.4 visualizes the first 6 hypersurfaces. This figure was directly adopted from Figure 2 of [19]. On the other hand, the polynomials of the parameters violating our assumptions are as follows:*

$$\beta_{1,2|\emptyset}^{(1)} - \beta_{1,2|\emptyset}^{(2)} : \quad B_{12}^{(1)} - B_{12}^{(2)} = 0,$$

$$\beta_{2,1|\emptyset}^{(1)} - \beta_{2,1|\emptyset}^{(2)} : \quad \frac{B_{12}^{(1)}\sigma_1^2}{(B_{12}^{(1)})^2\sigma_1^2 + \sigma_2^2} - \frac{B_{12}^{(2)}\sigma_1^2}{(B_{12}^{(2)})^2\sigma_1^2 + \sigma_2^2} = 0,$$

$$(\sigma_{2|\emptyset}^{(1)})^2 - (\sigma_{2|\emptyset}^{(2)})^2 : \quad (B_{12}^{(1)})^2\sigma_1^2 - (B_{12}^{(2)})^2\sigma_1^2 = 0,$$

$$(\sigma_{1|\{2\}}^{(1)})^2 - (\sigma_{1|\{2\}}^{(2)})^2 : \quad \frac{1}{(B_{12}^{(1)})^2\sigma_2^{-2} + \sigma_1^{-2}} - \frac{1}{(B_{12}^{(2)})^2\sigma_2^{-2} + \sigma_1^{-2}} = 0.$$

*Clearly, the number of polynomials that violate Assumptions 4.1 and 4.2 is much smaller as compared to those of the faithfulness assumption. HAs a consequence our assumption is weaker than the faithfulness assumption in the finite sample regime (where violations correspond to points that are close to any of the hypersurfaces).*

## E  Constraint-based method for estimating the difference-UG

In this section, we present a constraint-based method for estimating the difference-UG model in linear SEMs with general additive noise, i.e., where the noise is not necessarily Gaussian. Our constraint-based method is built on performing a hypothesis test on each $(i,j)$-th entry and then finding the set of $(i,j)$-th entries where $\Theta_{ij}^{(1)} \neq \Theta_{ij}^{(2)}$. The test for invariance of diagonal entries, i.e., $\Theta_{ii}^{(k)}$, is equivalent to the hypothesis test $H_0^{i|[p]\backslash\{i\}}$ as discussed in Section 3, since $(\sigma_{i|[p]\backslash\{i\}}^{(k)})^2 = (\Theta_{ii}^{(k)})^{-1}$. For the non-diagonal entries, since the non-zero pattern of $\Theta_{ij}^{(k)}$ is the same as the non-zero pattern of the partial correlation coefficients, i.e., $\rho_{ij|[p]\backslash\{i,j\}}^{(k)}$, we first find the set of non-diagonal entries that are different between $\Theta^{(1)}$ and $\Theta^{(2)}$ by doing partial correlation tests for each distribution and then comparing the non-zero patterns. After that, for each entry $(i,j)$ that is estimated to be non-zero in both $\Theta^{(1)}$ and $\Theta^{(2)}$, we use the test statistic:

$$\hat{Q} := \left(\hat{\Theta}_{ij}^{(1)} - \hat{\Theta}_{ij}^{(2)}\right)^2 \cdot$$
$$\left(\frac{\hat{\Theta}_{ii}^{(1)}\hat{\Theta}_{jj}^{(1)} + (\hat{\Theta}_{ij}^{(1)})^2}{n_1} + \frac{\hat{\Theta}_{ii}^{(2)}\hat{\Theta}_{jj}^{(2)} + (\hat{\Theta}_{ij}^{(2)})^2}{n_2}\right)^{-1}$$

and test if it fits the F-distribution with parameters $F(1, n_1 + n_2 - 2p + 2)$. If this is the case, we conclude that this particular entry $(i,j)$ is invariant between the two precision matrices. The consistency guarantees of $H_0^{i|[p]\backslash\{i\}}$ and partial correlation tests follow trivially from previous results. For $\hat{Q}$, it follows from Proposition 3 of [5] on the asymptotic normal distribution of the empirical precision matrix $\hat{\Theta}$ that if the null hypothesis is true, then $\hat{Q}$ converges in distribution to $\chi^2(1)$ as $n_1, n_2 \to \infty$.

## F  Additional high-dimensional evaluation

**High-dimensional setting: 10% changes.** We present the results of increasing the number of changes between the two DAGs, and hence the size of $S_\Theta$. We used the same simulation parameters as for Figure 2, i.e. $p = 100$ nodes, a neighbourhood size of $s = 10$, and sample size $n = 300$, except that the total number of changes was 10% of the number of edges in $B^{(1)}$, rather than 5%. As shown in Figure F.1, both initializations of the DCI algorithm still outperform separate estimation by GES and the PC algorithm. However, because the underlying DAGs have maintained constant sparsity while the difference-DAG has become more dense, the gains in performance by using the DCI algorithm have slightly diminished, as expected by our theoretical analysis.

(a) difference-DAG skeleton $\bar{\Delta}$

(b) difference-DAG $\Delta$

Figure F.1: ROC curves for estimating the difference-DAG $\Delta$ and its skeleton $\bar{\Delta}$ with $p = 100$ nodes, expected neighbourhood size $s = 10$, $n = 300$ samples, and 10% percent change between DAGs.

## G    Real data analysis - ovarian cancer

We tested our method on an ovarian cancer data set [18]. This data set consists of the gene expression data of patients with ovarian cancer. The patients are divided into six subtypes (C1-C6). The C1 subtype was characterized by differential expression of genes associated with stromal and immune cell types and is associated with shorter survival rates. In this experiment, we divide the subjects into two groups, group 1 with $n_1 = 78$ subjects containing patients with C1 subtype, and group 2 with $n_2 = 113$ subjects containing patients with C2-C6 subtypes. In this work, we focused on two pathways from the KEGG database [9, 15], the apoptosis pathway containing 87 genes, and the TGF-$\beta$ pathway with 82 genes.

We compared our results to those obtained by the DPM method [21], which infers the difference in the undirected setting. As input to Algorithm 2, we took $S_\Theta$ to be all of the nodes in the output of the DPM algorithm and took $\Delta_\Theta$ to be the fully connected graph on $S_\Theta$. We then learned the difference DAG using Algorithm 3. The final set of edges over different tuning parameters was chosen using stability selection as proposed in [13] and is shown in Figure G.1. This procedure identified two hub nodes in the apoptosis pathway: BIRC3 and PRKAR2B. BIRC3 has been shown to be an inhibitor of apoptosis [7] and is one of the top disregulated genes in ovarian cancer [8]. This gene has also been recovered by the DPM method as one of the hub nodes. While BIRC3 has high in-degree, hub gene PRKAR2B has high out-degree, making it a better candidate for possible interventions in ovarian cancer since knocking out a gene with high out-degree will have widespread downstream effects on the target genes. Indeed, PRKAR2B is a known important regulatory unit for cancer cell growth [2] and the RII-$\beta$ protein encoded by PRKAR2B has already been studied as a therapeutic target for cancer therapy [14, 3]. In addition, PRKAR2B has also been shown to play an important role in

(a) Apoptosis pathway

(b) TGF-$\beta$ pathway

Figure G.1: Estimate of the difference DAG between the two groups for the apoptosis and TGF-$\beta$ pathways. The black lines represent the edges discovered by both our method and DPM, the red lines represent the edges discovered only by our method, and the grey lines represent the undirected edges discovered only by DPM.

(a) apoptosis, PC          (b)TGF-$\beta$, GES          (c) TGF-$\beta$, PC

Figure G.2: Estimate of the difference DAG between the two groups for the apoptosis and TGF-$\beta$ pathways using the PC and GES algorithms.

disease progression in ovarian cancer cells [1]. Since the DPM method does not infer directionality, it is not possible to tell which of the hub genes might be a better interventional target. This is remedied by our method and its impact for identifying possible therapeutic targets in real data is showcased by finding an already known drug target for cancer.

For the TGF-$\beta$ pathway, our analysis identified THBS2 and COMP as hub nodes. Both of these genes have been implicated in resistance to chemotherapy in epithelial ovarian cancer [12], confirming the importance of our findings. These nodes were also recovered by DPM.

Overall, the undirected graph discovered by DPM is similar to the DAG found by our method. The disparity in the TGF-$\beta$ pathway between the difference UG model $\Delta_\Theta$ and the difference DAG model $\Delta$ can be explained by the fact that the edge between COMP−BMP7 in $\Delta_\Theta$ can be accounted for by the two edges BMP7→ID1 and COMP→ID1 in $\Delta$. Though these edges might represent the true regulatory pathways, the sparsity-inducing penalty in the DPM algorithm could remove them while leaving the edge between COMP and BMP7. This disparity between the two algorithms highlights the importance of replacing correlative reasoning with causal reasoning, and accentuates the significance of our contribution.

We also applied the GES and PC algorithms on the ovarian cancer data set. We considered the set of edges that appeared in one estimated skeleton but disappeared in the other as the estimated skeleton of the D-DAG $\bar{\Delta}$. In determining orientations, we considered the arrows that were directed in one estimated CP-DAG but disappeared in the other as the estimated set of directed arrows. Figure G.2 shows the results by applying the PC algorithm on the apoptosis and TGF-$\beta$ pathway and the results by applying GES on the TGF-$\beta$ pathway. Here we omitted GES results on the apoptosis pathway since GES algorithm did not discover any differences on the apoptosis pathway. Figure G.2 shows that PC and GES cannot discover any hub nodes.

(a) GES

(b) PC

Figure G.3: Estimate of the difference DAG between naive and activated T cells using the PC and GES algorithms.

## H   Real data analysis - T cell activation

We compare DCI with the GES and PC algorithms on the T cell activation data set. Figure G.3 (a) shows the results of applying GES to naive and activated data sets separately and calculating the difference. Figure G.3 (b) shows the estimated results of applying PC to the T cell data set.