[Reviews · NeurIPS 2018]

Reviewer 1



This paper introduces an algorithm for estimating the difference between two DAGs (dDAG) in the Gaussian-linear case. Importantly, the algorithm does not approach this by first learning the two DAGs separately, but instead relies on novel ideas to construct the dDAG directly. This work draws on existing work in two areas: (1) estimating the difference between two undirected graphs (UGs); (2) estimates the invariant parts of two DAGs (the complementary problem to the one proposed here). These ingredients are combined with several novel ideas to obtain the dDAG estimation algorithm. Strong aspects of the paper are: - The problem is well motivated. While at first the idea of having two different DAGs representing the same causal system appears slightly counterintuitive, in the context of linear models this makes sense of one thinks about it in terms of effect modification (where the magnitudes of structural ceofficients are affected by an intervention). - The presented algorithm is theoretically well developed and is convincingly shown to outperform other methods on simulated data. The ideas used in the algorithm appear novel to me and might be useful in other settings. Different versions of the algorithm are presented for high-dimensional and low-dimensional settings. The algorithm also does not try to "reinvent the weel" but makes good use of existing techniques, such as KLIEP, where this is appropriate. - There appears to be clear potential for this method to be useful in practice, even if the existing empirical analysis on "real data" is not very convincing yet (see below). - The paper is well-written and clear (but see a few minor issues below), and also the supporting information is of very high quality. I see two main points of criticism: - The approach seems to have some limitations that are not sufficiently discussed. First, there is the assumption that the two DAGs that are compared share the same topological order. This limitation should be stated more prominently, e.g., in the abstract, so users can decide immediately whether this can be justified for their application. As the authors mention, in biological networks, if one is willing to assume acyclicity in the first place then this assumption does not seem to be much more restrictive, but others might argue that already the acyclicity assumption is unrealistic to begin with. Second, the approach seems to have exponential time complexity because all subsets of S_\theta need ot be repeatedly considered, and some discussion of how this constrains applicability would be appreciated. - The empirical part appears less convincing than the theoretical part. The authors look at the learn dDAGs and then give literature references for roles of the transcripts they find to be "hub genes", but this appears to be somewhat post-hoc. What would the goal of such an analysis be? Ideally, that goal should be pre-specified, and only then should the analysis be carried out. It is not suprising to find literature references claiming a function for many of the transcripts; such references will likely be available for most transcripts that are differentially expressed. Specifically for the T cell data, the authors first focus on only the transcripts with a >10 fold change, which I would expect to lead to strong violations of the causal Markov assumption, since many important transcripts are now unobserved. Regardless, the authors present the finding that Granzyme B (GZMB) emerges as a root note with 3 children as a validation of the method. However, Interferon Gamma (IFNG) appears as a sink node with no children. This appears counterintuitive since both GZMB and IFNG are secreted by T cells upon activation, so they are end products of the activation process. As such the position of IFNG makes more sense to me, but in any case it shouldn't differ so substantially from GZMB. In summary, these analyses do prove that we can use the algorithms on real data but I would be careful with any biological interpretation at this point. Any sensible interpretation would require a more careful analysis of the findings. These issues nonewithstanding, I think this is a strong paper that makes an important contribution to the field. Minor points / requests for clarification: - Some of the symbols used in the notation are not very intuitive, such as the Delta with a bar for the skeleton and the A tilde for the arrows. - In Algorithm 1, perhaps slightly rephrase to clarify that once Delta_\Theta is estimated, S_\Theta can just be read off from that. The way it is written, it suggests that the two objects would both need to be estimated. - I believe that there is a typo in Algorithm 3 when you use the beta's instead of the sigma's . If that is not a typo, then I completely failed to understand what you mean in the explanation that starts at line 143. - In Example 3.3., try making more space between the arrowheads and the numbers, it looks a bit as if there was an arrow pointing to 1. - In line 161, I did not understand what you mean without consulting the supplementary material. After having a look at that, it appears to me that what you want to say is that some additional edges can be oriented given the requirement for acyclicity (which has nothing to do with v-structures per se). But then I don't see why this would be so computationally expensive. Why would you need to enumerate all directed paths (chains) to find one from x -- y? Is it not sufficient to perform a simple reachability query, which can be done in one graph traversal? - Line 202, for consistency, would it be better to write \sigma_i(k) instead of "the variance of \epsilon_i(k)" here? = After author response = Thank you for responding to my points and for agreeing to clarifying the paper in some respects.

Reviewer 2



The goal of this paper is to construct a DAG that represents the differences between two different but related causal DAG models (representing different causal relations in different contexts) from i.i.d. data. The difference graph has an edge from A to B when the coefficient for the edge is different in the two causal DAGs (including the possibility that the coefficient is zero in one of the DAGs.) This problem has been considered for undirected graphs, but not for directed graphs. For the purpose of planning experiments, directed graphs are essential however. The proposed algorithm (Difference Causal Inference or DCI) can be useful in biological problems where in different contexts some edges might have been removed or had their coefficients changed. DCI first constructs the adjacencies in the difference DAG by searching for regression coefficients that are invariant across the different contexts, and then identifies the direction of (some) of the edges by searching for regression coefficients that are invariant across the contexts. The obvious alternative to the proposed algorithm is to use standard algorithms such as PC or GES to construct both DAGs from the two different data sets, and then take their difference. However, this naive algorithm only finds edges that occur in only one DAG, rather than edges that appear in both DAGs but differ in their linear coefficients. In addition, because the difference DAG is often much sparser than the two original DAGs in different contexts, there are a number of statistical advantages to estimating the difference directly, as opposed to calculating the difference from estimates of the two original DAGs. (There has been similar work on inferring parts of two DAGs that are invariant across contexts, but it is not possible to infer the difference DAG from these invariants unless the two original DAGs are known.) One limitation of the algorithm proposed in the paper is that it makes a large number of strong assumptions: 1. Linearity 2. Gaussianity (which can be weakened at the expense of computational complexity). 3. The two causal DAGs have a common super-DAG (i.e. there are no edge reversals). 4. No latent confounders (which can be weakened to no changes in the latent confounder connections). 5. All differences in linear coefficients between the two DAGs entail differences in a certain set of regression coefficients. 6. All differences in noise terms between the two DAGs entail differences in a certain set of conditional variances. The last two assumptions are similar in spirit to the commonly made faithfulness assumption, in that they assume that certain polynomials are not equal to zero unless they are identically zero for all parameter values. However, they neither entail faithfulness nor are entailed by faithfulness. In terms of comparing the naive algorithms versus DCI, the paper makes a solid case that where their set of assumptions is applicable, DCI widely outperforms the naive algorithms. However, the paper should mention that these there are conditions under which the naive algorithms based on PC and GES algorithms can be applied where DCI cannot. For example, both of PC and GES can be applied to much larger numbers of variables than DCI can, and PC can be applied to a much wider class of distributions, while modifications of PC drop the assumption of no latent variables. The paper argues that assumptions 5 and 6 are generally weaker than the faithfulness assumption because the surfaces in the parameter space where the assumptions are violated are fewer in number for assumptions 5 and 6 than for the Faithfulness Assumption. However, the surfaces violating assumptions 5 and 6 are not a subset of the surfaces where faithfulness is violated. Both assumptions 5 and 6 on the one hand and faithfulness on the other hand have surfaces where they are violated of Lebesgue measure 0. If (as is more statistically relevant) one considers the volume of space close to a surface where the assumptions are violated, that depends on the shape of the surface as well as the number of surfaces. So while it seems like a reasonable conjecture that assumptions 5 and 6 are weaker than the faithfulness assumption, it seems like just a conjecture at this point. The paper notes that the standard version of the DCI algorithm does not have a complete set of orientation rules, since it does not orient edges to prevent cycles to reduce the computational complexity of the algorithm. The article states this is "computationally intractable in the presence of long paths." However, it is not clear why adding this rule would be a computational bottleneck for the algorithm or why the length of paths matter. It would seem that this would only require calculating the transitive closure of the directed part of the graph, which should be no worse than n^3, and could certainly be performed for hundreds of variables, which is in the range that the article applies the algorithm to. DCI was tested on both simulated and real data. The simulations used both 10 node DAGs and 100 node DAGs. In both cases, DCI significantly outperformed the naive algorithms. My only criticism of these tests is that they made all of the strong assumptions presupposed by DCI. This is quite commonly done, but given that DCI is applied to data where many of these assumptions are certainly false, it would be instructive to see how much violations of these assumptions affect the results, and the comparison with other algorithms. DCI was also applied to several real data sets, including one where there were two groups of ovarian cancer patients with different survival rates, and on where there were activated naive T cells. In each case the results were compared to an algorithm that computed a difference graph for undirected graphs. (It is not clear why the results were not also compared to the naive algorithms based on GES or PC). In each case they found nodes of importance (e.g. hub nodes) that were found to play an important role in the external literature. It is not clear from the exposition whether they found any part of the output surprising or likely to be wrong, or how systematically the output was compared to what was already in the literature. Overall, the article was written quite clearly. The general approach of DCI is similar to other algorithms which construct graphs representing edges that are invariant across different contexts (rather than different across different contexts) and also uses algorithms for constructing difference graphs for undirected edges. However, the particular application is both novel and significant, and both simulation tests and applications to real data show promise.

Reviewer 3



The paper proposes a method for learning a D-DAG, or a set of directed and undirected edges which represent causal relationships which are different across two datasets (in magnitude or existence), where it can be assumed that the causal relationships have the same topical ordering across the datasets and linear Gaussian models apply. Detecting changes in genetic regulatory networks is given as a motivating example. Existing methods are used to first estimate the undirected graph of differences (or the complete graph is used) and then a PC style approach with invariance tests using regression coefficients and residual variances are used to remove and orient edges. The method is shown to be consistent under some assumptions and evaluated using synthetic data and genetic regulatory network data. The paper address an interesting and relevant problem that, to my knowledge, has thus far remained unaddressed. The proposed method draws on previous related approaches but contains significant novel contributions. The general approach appears theoretically sound assuming the theorems in the supplement are correct (I was not able to check). There are some parts of the paper that are confusing and could be made more clear. At one point the paper seems to indicate that using the complete graph instead of KLIEP as input to algorithm 2 is for consistency purposes, but then later suggests both methods can be proved consistent. In section 2, paper says learning the DAGs separately and taking the difference cant identify changes in edge weights "since the DAGs are not identifiable" - I assume this just refers to the fact that in addition to learning the graph structures, you would need to regress each variable on its parents to find differences in edge weights but if this is what is meant, I'm not sure what "the DAGs are not identifiable" then refers to. Corollary 4.5 sounds like a completeness result (assuming I'm interpretting 'the D-DAG \Delta' correctly). The preceding text says it follows directly from Theorem 4.4 (orientation consistency). In general, this would not be the case. Some discussion explaining why this is the case in the setup of the paper would be clarifying. The assumption that both DAGs have the same topological order has implications both for the significance of this approach since it may be applicable to fewer problems, but also for clarity: effectively, the approach is for estimating differences in edge weights, but where it is assumed that the causal structure is the same (only some edges disappear as their weights go to zero), but the impression given from the abstract and initial description is that this method is for learning actual differences in the causal structure (not just edges that go to zero). This is a fundamentally different and harder problem. I have some concerns about the empirical results. The metric 'consistenty estimated D-DAGs' is not defined explicity, but I assume the use of 'consistency' indicates this metric is true whenever all of the edges in the D-DAG correspond to differences between the two graphs - in other words, it incorporates precision, but not recall. If this is the case, I have two concerns: 1) if it counts cases correctly where the D-DAG does not contain some edges that are different in the two graphs, then the increase in performance over PC/GES could be due to DCI being more conservative rather than more accurate; 2) only counting cases correctly where all of the edges in the D-DAG are 'consistently estimated' may not be telling the full story, e.g. it's possible that many of the cases that PC/GES did not consistently estimate were only slightly off, which might make the difference in average performance less dramatic - it would be interesting to see what mean/median precision and recall look like for the different methods. Additionally, the lowest sample size tested in the simulations is 1000, which seems high for many applications (the real data tested on only includes 300 samples). Why were the simulations not done at sample sizes more in range with the real data? The obvious worry is that the improvement over PC/GES may not be as significant at lower sample sizes.